# *Laminaria japonica* Peptides Suppress Liver Cancer by Inducing Apoptosis: Possible Signaling Pathways and Mechanism

**DOI:** 10.3390/md20110704

**Published:** 2022-11-10

**Authors:** Yingzi Wu, Yuanhui Li, Wenhai Guo, Jie Liu, Weiguo Lao, Penghui Hu, Yiguang Lin, Hongjie Chen

**Affiliations:** 1Department of Traditional Chinese Medicine, The Third Affiliated Hospital of Sun Yat-sen University, Guangzhou 510630, China; 2School of Chinese Medicine, Hong Kong Baptist University, Hong Kong 999077, China; 3National Marketing Center, Sinopharm Group Pharmaceutical Co., Ltd., Guangzhou 510010, China; 4State Key Laboratory of Respiratory Disease for Allergy and Shenzhen Key Laboratory of Allergy & Immunology, Shenzhen University School of Medicine, Shenzhen 518060, China; 5Department of Biochemistry, Douglass Hanly Moir Pathology, Macquarie Park, NSW 2113, Australia; 6Department of Oncology, Jiangmen Central Hospital, Jiangmen 529030, China; 7School of Life Sciences, University of Technology Sydney, Broadway, NSW 2007, Australia

**Keywords:** seaweed, *Laminaria japonica*, *Laminaria japonica* peptides, cancer, liver, cell cycle, apoptosis

## Abstract

The anticancer properties of *Laminaria japonica* peptides (LJPs) have never been studied. Here, we extracted LJPs from fresh seaweed and explored their anti-liver cancer activity (in vivo and in vitro). LJPs were isolated/purified by HPLC-ESI-MS. HepG2 cell apoptosis and cell cycle were evaluated. MTT assays were used to examine the cytotoxicity of LJPs. Caspase activation of caspases 3 and 9, cleaved caspases 3 and 9, and cleaved PARP was examined by Western blotting. The PI3K/AKT pathway and the phosphorylation states of MAPKs (p38 and JNK) were examined. We found that the LJP-1 peptide had the most antiproliferative activity in H22 cells in vitro. LJP-1 blocked H22 cells in the G0/G1 phase, accompanied by inhibition of cyclin expression. LJP-1 induced apoptosis through caspase activation and regulation of the ASK1/MAPK pathway. Concurrent in vivo studies demonstrated that LJP-1 significantly inhibited tumor growth and induced tumor cell apoptosis/necrosis. In conclusion, LJPs, particularly LJP-1, exert strong inhibitory effects on liver cancer growth in vivo and in vitro. LJP-1 induces HCC cell apoptosis through the caspase-dependent pathway and G0/G1 arrest. LJP-1 induces caspase-dependent apoptosis, in part by inhibiting PI3K, MAPK signaling pathways, and cell cycle proteins. LJP-1 has the potential to be a novel candidate for human liver cancer therapeutics.

## 1. Introduction

Hepatocellular carcinoma (HCC) is one of the most common forms of primary hepatic carcinoma and represents between 85% and 90% of primary liver cancers worldwide [1,2,3]. The incidence of HCC has almost tripled in recent decades. HCC poses a particularly serious sociomedical problem in Asia and sub-Saharan Africa, where the number of deaths is almost equal to the number of cases diagnosed annually (approximately 600,000), and the 5-year survival rate is below 9% [2,4]. Basic treatments for HCC include surgical resection, liver transplantation, percutaneous ablation, chemotherapy, and targeted therapy. However, the cure rate for patients who undergo resection is relatively low, and among patients who are not eligible for surgical or percutaneous procedures, only chemoembolization improves survival. Locally applied external radiation therapy, alone or in combination with chemotherapy, frequently leads to a significant improvement in primary and metastatic HCC [5,6], with a palliative effect in metastatic disease and possibly playing an equally palliative role in unresectable primary liver tumors [5]. Therefore, the development of new effective and more therapeutic agents remains urgently needed for the treatment of HCC.

Natural products have been and are currently a valuable source of anticancer drugs in drug discovery [6,7]. It is widely accepted that intricate architectures generally composed of natural products offer numerous opportunities and clues for the exploration of novel drugs. Recent studies have demonstrated that seaweed contains various bioactive components that have strong anticancer properties [8,9,10,11,12,13,14,15]. Various edible seaweed extracts have been shown to have protective effects against some types of cancer, demonstrating suppressive effects against chemically induced tumorigenesis through suppression in the initiation and promotion phases [16]. *Laminaria japonica* (*L. japonica*), a brown seaweed, is one of the most important seaweeds and has been widely used as a health food and a traditional oriental herbal medicine for more than a thousand years [17,18,19]. Many studies have shown that fucoidan, a complex polysaccharide extracted from brown seaweed, has promising anticancer effects against multiple types of cancer through various anticancer mechanisms, such as cell cycle arrest, apoptosis evocation, and stimulation of cytotoxic natural killer cells and macrophages [12,13,15]. In addition to its anticancer properties, fucoidan possess other pharmacological properties, including antioxidant, anticoagulant, anti-inflammatory, and immunomodulatory activities [20,21,22]. Fucoxanthin, another major active component extracted from *L. japonica*, has recently been reported to have an anti-lung cancer effect [19] and anti-liver cancer activities [9]. Importantly, a novel glycoprotein isolated from *L. japonica* has been shown to stimulate normal gastrointestinal cell growth by activating the epidermal growth factor receptor (EGFR) signaling pathway [23]. On the contrary, the isolated glycoprotein inhibits colon cancer cell proliferation, demonstrated in HT-29 cells, by inducing cell cycle arrest and apoptosis [4].

Research is currently being conducted on polysaccharides (such as fucoidan) and the main carotenoids (fucoxanthin) derived from *L. japonica,* as stated above. However, studies on the protein and peptide moieties present in *L. japonica* are limited, despite the fact that *L. japonica* is composed of more than 10% proteins and peptides [24]. No studies investigating the anticancer properties of *L. japonica* peptides have been reported to date. Therefore, in the present study, we first isolated the peptides from *L. japonica* and then screened and investigated the anticancer effects and possible underlying anticancer mechanisms of the *L. japonica* peptides through in vivo and in vitro experiments using various human liver cancer cell lines and a liver cancer animal model.

## 2. Results

### 2.1. Isolation and Analysis of LJPs

In the current study, 158 g of total protein was isolated from *L. japonica*. Then, ultrafiltration, hydrophobic chromatography, anion exchange chromatography, gel filtration chromatography, and RP-HPLC were performed in conjunction with anticancer activity tracking detection. Finally, four pentapeptides, seven hexapeptides, and a heptapeptide were isolated, and the amino acid sequence was identified (Table 1). After the LJPs were obtained, HPLC-ESI-MS and Edman degradation experiments were carried out to analyze the amino acid sequence of the LJPs.

LJP-1 was analyzed using HPLC-ESI-MS for molecular mass determination and peptide characterization. As shown in Figure 1, the fragment of ion m/z 602.2921 was considered a fragment [M+H]^+^. The fragment of ion m/z 471.1991 was considered the y4 ion, whereas m/z 425.1929 was considered the [b4-COOH+H]^+^ fragment, m/z 416.2296 was considered the b3 ion, m/z 285.1349 was considered the b3-b1 ion, and m/z 269.1602 was considered b2. The ion (m/z 251.1500) was considered the [b2-H_2_O+H]^+^ fragment, the ion (m/z 223.1553) was the [b2-COOH+H]^+^ fragment, m/z 156.0767 was the [His-H_2_O+H]^+^ fragment, and m/z 110.0705 was the [His-COOH+H]^+^ fragment. Based on HPLC-ESI-MS data and corresponding by Edman degradation results, we concluded that the LJP-1 sequence was EGFHL. The rest of the LJPs were identified using the same method as LJP-1; their amino acid sequences are listed in Table 1.

### 2.2. LJPs Inhibited HCC Cell Growth In Vitro

Three representative liver cancer cell lines, HuH7, HepG2, and H22, were specifically chosen to explore the effects of LJPs in in vitro liver cancer studies. As shown in Table 1, six LJPs (LJP-1, LJP-4, LJP-8, LJP-9, LJP-11, and LJP-12) exhibited significant antiproliferative activity against the three liver cancer cell lines. LJP-1 showed significant antiproliferative activity in HuH7, HepG2, and H22 cell lines, with IC_50_ values of 0.48 ± 0.05 mM, 0.45 ± 0.05 mM, and 0.36 ± 0.03 mM, respectively. Among the cell lines tested, the H22 cell line had the lowest IC_50_ value and was therefore chosen for further experiments. The results indicated that LJP-1 showed a wide range of growth-inhibitory activity against liver cancer cells.

With promising results showing antiproliferative activity of LJP-1 in the three liver cancer cell lines tested, we then investigated cytotoxicity in normal cells using the same technique. We found that the cytotoxicity of LJP-1 in normal cells was significantly less than in cancer cells. Cytotoxicity, or killing effect, in the human liver cell line HL-7702 required much higher doses of LJP-1 (IC_50_ value of 7.76 ± 0.71 mM for HL-7702) compared to tumor cells, suggesting that LJP-1 was a suitable candidate as an anticancer drug for further study.

### 2.3. LJP-1 Induced Cell Cycle Arrest by Regulating Cyclin Expression

To investigate the anticancer activity of LJP-1 in cell cycle arrest, we measured the cell cycle phases of H22 cells by flow cytometry using propidium iodide (PI) staining after LJP-1 treatments. As shown in Figure 2A, H22 cells in phase G0/G1 were significantly blocked by 38.8% (control), 44.3% (0.36 mM, vs. control, * *p* < 0.05), and 56.1% (1.80 mM, vs. control, * *p* < 0.05).

We analyzed the expression of proteins involved in cell cycle regulation to examine the molecular events of LJP-1 inhibition of the G0/G1 transition; the results are shown in Figure 2B. LJP-1 suppressed the expression of cyclin D and cyclin E and attenuated the expression of both CDK2 and CDK6. Furthermore, the expression of both p21 and p27 was upregulated by LJP-1 treatments, indicating that LJP-1 is related to cell cycle arrest and dependent on p21 and p27. In particular, these results indicate that LJP-1 modulates the G0/G1 phase proteins, resulting in cycle arrest.

### 2.4. LJP-1 Induced Caspase-Dependent Apoptosis in H22 Cells

To investigate whether LJP-1 induced arrest in the G0/G1 phase in H22 cells through the apoptotic pathway, we first performed an annexin V-FITC/PI double-staining assay. As shown in Figure 3A, LJP-1 induced significant apoptosis in H22 cells. After LJP-1 treatments, the percentage of apoptotic cells increased from 3.42% (Control) to 21.0% (0.36 mM, * *p* < 0.05) and 34.0% (1.80 mM, * *p* < 0.05).

In the experiment examining whether caspase activation is involved in LJP-1-induced cell apoptosis, we found that, as shown in Figure 3B, LJP-1 increased the expression of cleaved caspase-9 and cleaved caspase-3. The cleaved poly-ADP ribose polymerase (PARP) was also strongly activated. The above results suggest that LJP-1 induces programmed apoptosis by activating the caspase cell death pathway.

### 2.5. LJP-1 Induced Apoptosis of H22 Cells by Regulating p38-MAPK and PI3K/AKT Pathways

The PI3K/AKT pathway acts as a key regulator of cellular survival and apoptosis in most cancers. We therefore evaluated whether PI3K/AKT played a central role in LJP-1-mediated apoptotic cell death. LJP-1 treatments (both LJP-1-L and LJP-1-H) significantly downregulated PI3K expression and increased p-AKT expression (Figure 4A), indicating that the PI3K/AKT pathway also played a key role in LJP-1-mediated apoptosis. LJP-1 also decreased apoptosis signal-regulating kinase 1 (ASK1). Furthermore, we investigated whether LJP-1 treatments regulated MAPK phosphorylation states (p38 and JNK). As shown in Figure 4B, both p38 and JNK phosphorylation were activated in H22 cells after LJP-1 treatments.

### 2.6. LJP-1 Inhibited Tumor Growth In Vivo

Because LJP-1 exhibited significant in vitro anticancer activity in H22 cells, further experiments were carried out to investigate the in vivo antitumor effect of LJP-1. Tumors were induced in mice by injecting H22 cells. When tumor volume reached 300 mm^3^, mice were randomly divided into three groups as described in Section 2.5. The animals were treated with saline or LJP-1 for 3 weeks. To evaluate tumor growth in mice, tumor volume was calculated, and tumor weight was recorded (Figure 5).

As shown in Figure 5, tumor growth inhibition was observed in the LJP-1 treatment groups (both treated with 2 mg/kg and 10 mg/kg) compared to the control group. Tumor volume was significantly decreased in LJP-1-H treatment groups from day 15 after LJP-1 treatment. The magnitude of inhibition, measured as tumor volume, was very significant and up to 50% on day 19 and day 23 at a dose of 10 mg/kg.

### 2.7. LJP-1 Induced Cell Apoptosis and Tumore Necrosis

In the current experiment, cancer tissue was stained with ordinary TUNEL staining to detect whether LJP-1 treatment inhibited tumor growth through cell apoptosis. The results are shown in Figure 6A; the apoptotic cells are stained brown. Apoptotic cells in the LJP-1-H group were significantly higher than in the control group. The results showed that approximately 34% more apoptotic cells were observed in the LJP-1-H-treated group than in the control group. Tumor tissue was fixed with formalin, and paraffin sections were stained with H&E. As shown in Figure 6B, there were large necrotic cells in the tumor tissue, as well as pyknosis fragmentation and nuclei dissolution. The necrotic area of tumor tissue in the LJP-1-H-treated group was significantly greater than in the control group. Furthermore, the intercellular space between tissues was significantly widened. In the current experiment, the number of apoptotic cells was counted in six visual fields for each group. After TUNEL fluorescent staining, apoptotic cells were stained green and observed under a fluorescence microscope (Figure 6C). Compared to the control group, the results showed that LJP-1-H treatments induced significant cell apoptosis (*p* < 0.05).

## 3. Discussion

Previous studies have shown that isolated *L. japonica glycoprotein* inhibits cancer cell proliferation by inducing cell cycle arrest and apoptosis [25,26]. However, studies on the anticancer effect of *L. japonica* peptides have not been carried out. In the present study, we isolated *L. japonica* peptides (LJP-1 to LJP-12) using an activity-tracking method; subsequently, we screened the anticancer (HCC) properties of these peptides. We found that six LJPs (LJP-1, LJP-4, LJP-8, LJP-9, LJP-11, and LJP-12) exhibited significant antiproliferative activity against the three cancer cells tested. We then further investigated the anticancer properties of LJP-1 and the underlying mechanism using the H22 liver cancer cell line and an in vivo animal model. For the first time, we demonstrated that LJP-1 has strong inhibitory effects on liver cancer growth, significantly suppressing tumor growth and decreasing tumor volume by more than 50% at a dose of 10 mg/kg on day 23 compared to the control (Figure 5). Furthermore, we found that the underlying mechanism is associated with apoptosis of HCC cells through the caspase-dependent pathway and through phase arrest G0/G1 by regulating cell cycle checkpoint proteins. Furthermore, LJP-1 inhibited the PI3K/AKT and MAPK signaling pathways. 

Deregulation of cell cycle progression is a common feature of liver cancer cells. Therefore, targeting regulatory cyclins has been proposed as an important strategy for the treatment of human liver cancer [27,28]. In our in vitro experiments, H22 cells were shown to accumulate significantly in the G0/G1 phase. The cell cycle is regulated by multifaceted proteins, including two classes of molecules: cyclin-dependent kinases (CDK) and cyclin-binding partners. Cyclin D and cyclin E (along with CDK2 and CDK6) play a central role in the G0/G1 phase of the cell cycle [29]. Our studies demonstrated suppression of cyclin D and E expression and attenuation of CDK 2 and 6 expression in cells treated with LJP-1, suggesting that these proteins are involved in the LJP-1-induced cell inhibitory pathway in cancer. P21 and p27 are potent cyclin-dependent kinase inhibitors that bind to and inhibit the activities of CDKs; thus, increased expressions of these proteins indicate the induction of G0/G1 cell cycle arrest [30]. In the current study, we also demonstrated up-regulation of p21 and p27 expression after LJP-1 treatments, supporting that hypothesis that LJP-1 is related to p21- and p27-dependent cell cycle arrest. It should be noted that LJP-1 modulated the expression of the G0/G1 phase protein, resulting in cycle arrest.

The results of the cell cycle analysis revealed that LJP-1 arrested H22 cells in the G0/G1 phase, indicating the occurrence of apoptosis. The results of cell apoptosis were further confirmed by flow cytometry detection. Caspase activation is usually accompanied by PARP activation, which indicates activation of the DNA repair mechanism. Caspase activation is considered a hallmark of apoptosis [31,32]. In the current study, LJP-1 significantly increased the expressions of cleaved caspase-9, -3, and PARP was strongly activated (Figure 3B), suggesting that LJP-1 induced cell apoptosis through caspase activation, a mitochondrial apoptosis pathway.

The ASK1/MAPK pathway is involved in cell cycle regulation and apoptosis. A variety of stresses, such as ROS accumulation, can activate ASK1. ASK1 phosphorylation can lead to downstream activation of the p38 MAPK pathway [33]. Furthermore, the phosphatidylinositol-3-kinase (PI3K)/AKT pathways are common in human cancer, and there is increasing evidence that PI3K/AKT is involved in the development of many types of cancers. Previous studies suggested that PI3K/AKT and its downstream pathways are promising targets for therapeutic intervention [3,27,34].

An interesting finding of the current study is that we found that LJP-1 negatively regulated PI3K expression and simultaneously increased (not decreased) the expression of p-AKT (Figure 4A). How is AKT activation related to apoptosis activation and inhibition of tumor growth? One possible explanation is that activated AKT promotes autophagy, leading to autophagic cell death, resulting in tumor suppression and inhibition of tumor cell proliferation due to the dual effects (suppression and promotion) of autophagy in liver cancer [35]. The AKT/mTOR pathway is known to be a key regulator of autophagy, and AKT activation can stimulate downstream targets through the AKT/mTOR pathway; subsequent dephosphorylation of human homologues Unc-51-like autophagy-activating kinase-1 (ULK1) and Unc-51-like autophagy-activating kinase-2 (ULK2) leads to the promotion of autophagy [36]. Suppression of the mTOR pathway can also initiate autophagy [36]. Many studies have shown that autophagy functions as a tumor suppressor in liver cancer, and compromised autophagy could promote liver tumorigenesis [37]. Therefore, it is possible that the activation of AKT as a result of LJP-1 treatment can promote autophagy, leading to apoptosis/tumor suppression. We cannot confirm the molecular events that link AKT activation with the proposed autophagic cell death, as they are beyond the scope of this study. It would be interesting to conduct further studies investigating the molecular role of autophagy in controlling apoptosis in HCC.

Furthermore, LJP-1 activated p38 and JNK (MAPK phosphorylation states) of H22 cells, as shown in Figure 4B. These results indicate that LJP-1-induced H22 cell apoptosis may occur through the regulation of the MAPK pathway and possibly the PI3K/AKT pathway. Multiple apoptotic pathways work together through cross talk, leading to tumor suppression, since cross-talk of pathways is common in cancer signaling [35,37,38,39,40,41].

An important finding of our study was that LJP-1 exhibited very strong in vivo antitumor activity against HCC in tumor-bearing mice. Tumor volume was decreased significantly in animals treated with either a low dose of 2 mg/kg or a high dose of 10 mg/kg of LJP-1, as shown in Figure 5. The inhibitory effect was more prominent in the group treated with high doses, and a significant decrease was observed from day 15 after LJP-1 treatments, with a further decrease on days 19 and 23, indicating its potential therapeutic application for the treatment of human liver cancer. Mechanistically, we found that in our experiments, LJP-1 treatment significantly induced cancer cell apoptosis in liver cancer tissue (Figure 6), reaffirming that the underlying mechanism of the anticancer property of LJP is apoptosis-related.

Because no previous studies have been conducted on the anticancer effect of *L. japonica* peptides, the most relevant supporting data could be those from studies using the *L. japonica* glycoprotein (LJGP) and the *L japonica*-derived polysaccharide, fucoidan. Go et al. found that an LJGP from brown seaweed inhibited the proliferation of several cancer cell lines, including AGS, HepG2, and HT-29, in a dose-dependent manner by inducing apoptosis [26]. Furthermore, it increased cell growth in normal intestinal cells through the EGFR signaling pathway [23]. Mechanistically, they found that LJGP induced Akt/ERK activation and downregulated JNK/p38, which is similar to the findings of the current study using LJP-1. Increasing evidence supports the notion that fucoidan has strong antitumor actions, inhibiting the growth of various cancer cells by inducing cell cycle arrest and apoptosis and regulating growth-signaling molecules [12,13,15]. Duan et al. recently reported that fucoidan effectively suppressed HCC by inducing apoptosis through the p38 MAPK/ERK and PI3K/AKT signal pathways [10]. We demonstrated here that the PI3K/AKT and MARK pathways play a key role in LJP-1-mediated apoptosis. Our findings, together with previous work on LJGP and fucoidan, suggest that these bioactive components derived from *L japonica,* i.e., fucoidan, LJGP, and LJPs, share a similar mechanism of action in their anticancer activities, although they are structurally very different.

Our work provides original evidence to support the potential of LJP as a natural derivative of seaweed with significant anti-liver cancer activities, highlighting it potential for novel treatment of liver cancer. Our findings add another component derived from *L. japonica* to the list of anticancer bioactive substances derived seaweed. LJP shares a mechanism of action similar to that of fucoidan and LJGP in their anticancer actions. However, the link between LJPs, LJGP, and fucoidan is unknown. Current understanding the anticancer effect of LJPs is very limited. Furthermore, it would be interesting to explore the basic pharmacokinetics of LJP as an anticancer agent. Therefore, further work is needed to elucidate the potential of LJP as a new and effective natural therapy for liver cancer.

## 4. Materials and Methods

### 4.1. Materials and Cell Culture

Liver cancer cells (HuH7, HepG2, and H22) and human liver cells (HL-7702) were obtained from the Cell Bank of the Chinese Academy of Sciences (Shanghai, China). Fresh *L. japonica* was acquired from the South China Agricultural University (Guangzhou, China) and identified by Professor C. Wang (South China Agricultural University, Guangzhou, China). The annexin V-FITC/PI staining assay kit and propidium iodide (PI) reagents were supplied by Beyotime (Shanghai, China). Dulbecco’s modified Eagle medium (DMEM), fetal bovine serum (FBS), and penicillin-streptomycin (PS) were purchased from Gibco BRL (Life Technologies, Grand Island, NY, USA). The antibodies used in the current study were acquired from Cell Signaling Technology, Inc. (Manchester, NH, USA). Cell lines were grown in specific medium supplemented with 10% FBS. Cells were grown in a humidified atmosphere with 5% CO_2_ in incubators maintained at 37 °C. Balb/c mice (6–8 weeks old, 20 ± 2 g) were acquired from the Guangdong Experimental Animal Center.

### 4.2. Isolation and Amino Acid Sequence Analysis of LJPs

All isolation procedures were conducted at 4 °C. Fresh *L. japonica* (8 kg) was minced in a homogenate mixed with isopropanol in a ratio of 1:6 (*w*/*v*) and stirred uninterrupted for 4 h. Subsequently, the sediment was collected and lyophilized. The defatted precipitate (732 g) was dissolved (5%, *w*/*v*) in 0.50 M phosphate buffer solution (PBS, pH 7.5). After centrifugation (8000× *g*, 10 min), the supernatant was collected and freeze-dried as the total protein.

Total protein was fractionated using ultrafiltration with a membrane cutoff of 1 kDa molecular weight (MW) (Millipore, Wanchai, China). Two peptide fractions, LJP-A (MW < 1 kDa) and LJP-B (MW > 1 kDa), were collected and freeze-dried.

#### 4.2.1. Hydrophobic Chromatography

LJP-A was dissolved in 1.20 M (NH_4_)_2_SO_4_ prepared with 30 mM phosphate buffer (pH 7.5) and loaded onto a hydrophobic column of phenyl sepharose CL-4B (3.0 cm × 120 cm). A stepwise elution with decreasing concentrations of (NH_4_)_2_SO_4_ (1.20, 0.60, and 0 M) was dissolved in 30 mM phosphate buffer (pH 7.5) at a flow rate of 3.0 mL / min. Each 100 mL fraction was collected and monitored at 280 nm. Five fractions (LJP-A-1~LJP-A-5) were collected, and antiproliferative activity was detected against the three cancer cells. The fraction with significant anticancer activity was collected and prepared for anion-exchange chromatography.

#### 4.2.2. Anion-Exchange Chromatography

The LJP-A-3 fraction was collected (LJP-A-3, 6 mL, 4.42 g/mL) and injected into a DEAE-52 cellulose (Sigma-Aldrich, Shanghai, China) anion-exchange column (2.0 × 100 cm) equilibrated with deionized water and stepwise eluted with distilled water solutions of 0.30, 0.60, and 1.20 M (NH_4_) 2SO_4_ at a flow rate of 2.0 mL/min. Each eluted fraction (50 mL) was collected and monitored at 280 nm. Seven fractions (LJP-A-3-1~LJP-A-3-7) were collected, and antiproliferative activity was detected against the three cancer cells. The fraction with the highest antiproliferative activity was collected and prepared for gel filtration chromatography.

#### 4.2.3. Gel Filtration Chromatography

The LJP-A-3-5 fraction was collected and fractionated in a Sephadex G-25 column (Sigma-Aldrich, Shanghai, China) (2.0 × 100 cm) with a flow rate of 2.0 mL/min. Each eluate (50 mL) was collected and monitored at 280 nm. Five fractions (LJP-A-3-5-1~LJP-A-3-5-5) were collected, and antiproliferative activity against the three cancer cells was detected. The fraction with significant antiproliferative activity was collected and prepared for reversed-phase high-performance liquid chromatography (RP-HPLC).

#### 4.2.4. RP-HPLC and HPLC-ESI-MS Analysis

The LJP-A-3-5-3 fraction was collected. Subsequently, LJP-A-3-5-3 was separated by RP-HPLC (Agilent 1200) on a Zorbax SB C-18 column (4.6 × 250 mm, 5 µm). The elution solvent system was composed of water-trifluoroacetic acid (solvent A; 100:0.1, *v*/*v*) and acetonitrile-trifluoroacetic acid (solvent B; 100:0.1, *v*/*v*). The peptides were separated using a 30% to 70% gradient elution of solvent B for 45 min at a flow rate of 1.0 mL/min, with the detection wavelength set at 280 nm.

HPLC-ESI-MS was performed on a SCIEX X500R Q-TOF mass spectrometer (Framingham, MA, USA). The MS conditions were as follows: ESI-MS analysis was performed using a SCIEX X500R Q-TOF mass spectrometer equipped with an ESI source. The mass range was set to m/z 100–1200. The Q-TOF MS data were acquired in positive mode, and the MS analysis conditions were as follows: CAD gas flow rate, 7 L/min; drying gas temperature, 550 °C; ion spray voltage, 5500 V; declustering potential, 80 V; software-generated data file: SCIEX OS 1.0. The peptide is usually protonated under ESI-MS/MS conditions, and fragmentations occur mostly at the amide bonds because it is difficult to break the chemical bonds of the side chains at such low energy. Hence, on the basis of the HPLC-ESI-MS data and accompanied by the Edman degradation results, the amino acid sequences of the LJPs were identified.

#### 4.2.5. Summary of Key Steps

The key steps involved in the isolation of LJPs and the identification of amino acid sequences of LJPs are summarized in Figure 1.

### 4.3. Cell Apoptosis and Cell Cycle Assays

Ab MTT assay was used to detect the cytotoxicity activity of LJPs in vitro [42]. Briefly, 5 × 10^3^ cells/well were seeded in 96-well plates, and varying concentrations of LJP were added. After 48 h, the MTT solution (5 mg/mL) was added for an additional 4 h. The absorbance was measured at 570 nm using a microplate reader (Bio-Rad; Hercules, CA, USA) after 100 μL of DMSO was added. The IC^50^ values were determined by the non-linear multipurpose curve-fitting program.

After treatment with LJP-1 (0, 0.36, and 1.80 mM representing control, low dose of LJP-1 and LJP-1-L, and high dose of LJP-1 and LJP-1-H, respectively) for 48 h, the apoptosis of H22 cells was evaluated by flow cytometry. Briefly, after LJP-1 treatment, H22 cells were collected, washed with cold PBS, suspended in binding buffer (100 μL) (BD Biosciences, San Jose, CA, USA), treated with annexin V and propidium iodide (PI) (BD Biosciences), and incubated in the dark for 15 min. Then, another 300 μL binding buffer was added, and flow cytometry analysis was performed for 1 h to measure the rate of apoptosis. The percentages of cells in the G0/G1, S, and G2/M phases were determined using a cell cycle detection kit (BD Biosciences, Haryana, India) using a Beckman Coulter EPICS ALTRA II cytometer (Beckman Coulter, Brea, CA, USA).

### 4.4. Western Blot Analysis

After treatment with LJP-1 (0, 0.36, and 1.80 mM representing control, low dose of LJP-1 and LJP-1-L, and high dose of LJP-1 and LJP-1-H, respectively) for 48 h, H22 cells were collected, and protein was collected and measured with a Bradford protein assay to calculate the amount of protein. Equal amounts of protein were separated by sodium dodecyl sulfate polyacrylamide gel electrophoresis (SDS-PAGE) and electroblotted onto a polyvinylidene difluoride (PVDF) membrane. Immunoblots were blocked with 5% non-fat milk and incubated with primary antibody (1:1000) at 4 °C overnight, followed by incubation with the second conjugated peroxidase antibody (1:5000) at room temperature for 2 h [43]. Protein bands were measured, and *β*-actin was used as an internal standard of process control. The blot band densitometry was analyzed with ImageJ software, version 1.53t.

### 4.5. Immunization of Liver-Cancer-bearing Mice

The procedures for the animal study were approved by the Animal Ethics Committee of Shenzhen University. Mice were kept in a specific pathogen-free facility with free access to food and water. H22 cells were cultured and suspended in PBS, and 18 Balb/c mice were subcutaneously injected with H22 cells (1 × 10^6^) in the right back near the skin of the underarm. Tumor growth was measured every 2 days. When tumor volume reached 300 mm^3^, mice were randomly divided into three groups and treated with the following procedures: Group 1 (Control): mice were injected with 200 µL saline every 4 days around tumor tissue; Group 2 (LJP-1-L): mice were injected with LJP-1 (in 200 µL saline) 2 mg/kg every 4 days (4 days in a cycle, four times) around tumor tissue; Group 3 (LJP-1-H): mice were injected with LJP-1 (in 200 µL saline) 10 mg/kg every 4 days (4 days in a cycle, four times) around tumor tissue. All mice were sacrificed on day 23. The subcutaneous tumors were excised, and the tumor index was calculated. Furthermore, tumor tissues were cut into small woven pieces and fixed with polyformaldehyde, and paraffin sections were made later.

### 4.6. Evaluation of Tumor Apoptosis and Hematoxylin and Eosin (H&E) Staining

Apoptotic cells in cancer tissue sections were stained with a TUNEL reagent kit following the manufacturer’s instructions. Hematoxylin and eosin (H&E) staining was performed according to the routine procedure. The paraffin section was dewaxed and dehydrated with xylene and gradient alcohol, then stained with hematoxylin solution for 5 min, soaked in 1% acid ethanol (1% HCL in 70% ethanol) for 2 min, and rinsed in distilled water. Slices were stained with eosin solution for 3 min, dehydrated with graded alcohol, and cleaned with xylene.

### 4.7. Statistical Analysis

Data are presented as mean ± standard deviation (SD). GraphPad Prism 5.0 (Graph Pad Software, La Jolla, CA, USA) was used for statistical analysis. Statistical analysis was performed with one-way analysis of variance (ANOVA). *p* < 0.05 was considered statistically significant.

## 5. Conclusions

In conclusion, our study demonstrates that LJP-1 isolated from *L. japonica* possesses strong antitumor effects against HCC-based on evidence from both in vitro and in vivo studies. In vivo experiments show that LJP-1 strongly suppressed tumor growth on day 14 after treatment at doses of 2 and10 mg/kg. Our data supports that LJP-1 controls liver cancer proliferation by inducing HCC cell apoptosis through the caspase-dependent pathway and by arresting cells in the G0/G1 phase by regulating cell cycle checkpoint proteins. Furthermore, LJP-1 induces caspase-dependent apoptosis, in part, by inhibiting MAPK signaling pathways. The PI3K/AKT pathway and possibly other pathways may be involved. In addition, our screening study shows that other LJPs, namely LJP-4, LJP-8, LJP-9, LJP-11, and LJP-12, also exhibit antitumor activities. Therefore, *L. japonica peptides* (not just LJP-1) could be potential drug candidates for the novel treatment of human liver cancer and warrant further investigation.

## Data Availability

The data (figures and tables) used to support the findings of this study are included within the article.

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
