# Peer review of "Laminaria japonica Peptides Suppress Liver Cancer by Inducing Apoptosis: Possible Signaling Pathways and Mechanism"

_marinedrugs, 2022, doi:10.3390/md20110704_

Round 1
Reviewer 1 Report
This manuscript is well-written and suitable for publication in Marine Drugs after major revision.
1) What is the difference between LJP-L and LJP-H?
2) Figure 2A is too small to read.
3) The authors suggested that the PI3K/AKT pathway was inhibited following LJP-1 treatment. However, AKT was clearly increased in cells treated with LJP-1 (see Figure 4). As you know, the AKT is associated with cancer cell proliferation, growth, and survival. The authors should discuss the relationship between increased AKT activation and increased apoptosis in cells treated with LJP-1. Furthermore, the authors should revise the following sentences:
Line 30: inhibiting PI3K/AKT
Line 184: Figure 4 LJP-1 inhibited the PI3K/AKT pathway
4) Table 2 is the same one with Figure 5. Please delete one of them.
5) Figure 5: The authors should explain why the injection volume is different between LJP-L (2 mg/kg) and LJP-H (10 mg/ka). Furthermore, the number of mice used in the experiment should be added.
Author Response
Thank you very much for your constructive coments.
Our response is in the attached PDF file.

Reviewer 2 Report
This manuscript explores the anticancer properties of Laminaria japonica peptides (LJPs) in many liver cancer cell lines. They demonstrate the cytotoxicity activity of these peptides, inducing apoptosis as a principal molecular mechanism.
Even though this work is well structured and the large amount of experimental data shown, I have some doubts and comments.
Comments:
- In figure 2 the authors decided to use the H22 cell lines to analyze the cell cycle regulation induced by LJP-1, and all subsequent experiments. Knowing why the authors explicitly used this cell line will be helpful.
- It is desirable that the treatment times with the LJP-1 be evident in the legends of the different experiments (generally 48 hours).
- In point 2.5 the authors speak of changes in the expression of some proteins, which could be more precise. According to Western blot results, changes in protein levels can be seen, which may be due to several reasons, only one of them being the change in expression patterns. Please fix.
- Along the same lines, it is interesting to see that LJPs can induce a decrease in PI3K protein levels, the phosphorylation of one of its downstream targets (Akt) increases. Could other signaling pathways be involved in this event?
- In this work, it is clear that the LJPs protein can regulate the levels and degree of phosphorylation of some signaling proteins, which have an essential role in cancer. However, no data prove that these signaling pathways have a central role in the molecular mechanism of death induced by LJP1; moreover, the increase in Akt phosphorylation is contrary to what was expected since the activation of this protein is associated with the processes of cell survival and proliferation.
Given this last point, it is suggested to review the statement in this work's title.
Author Response

(The authors gave the same response as above.)

Round 2
Reviewer 1 Report
The manuscript has been revised adequately and I suggest that it is suitable for publication in Marine Drugs.